# Orthogonal Gradient Boosting for Interpretable Additive Rule Ensembles

## Abstract

Gradient boosting of decision rules is an efficient approach to find interpretable yet accurate machine learning models. However, in practice, interpretability requires to limit the number and size of the generated rules, and existing boosting variants are not designed for this purpose. Through their strict greedy approach, they can increase accuracy only by adding further rules, even when the same gains can be achieved, in a more interpretable form, by altering already discovered rules. Here we address this shortcoming by adopting a weight correction step in each boosting round to maximise the predictive gain per added rule. This leads to a new objective function for rule selection that, based on orthogonal projections, anticipates the subsequent weight correction. This approach does not only correctly approximate the ideal update of adding the risk gradient itself to the model, it also favours the inclusion of more general and thus shorter rules. Additionally, we derive a fast incremental algorithm for rule evaluation, as necessary to enable efficient single-rule optimisation through either the greedy or the branch-and-bound approach. As we demonstrate on a range of classification, regression, and Poisson regression tasks, the resulting rule learner significantly improves the comprehensibility/accuracy trade-off of the fitted ensemble. At the same time, it has comparable computational cost to previous branch-and-bound rule learners.

## 1 Introduction

Algorithms for learning additive rule ensembles (or rule sets) are an active area of research, because they are intrinsically interpretable yet relatively accurate due to their modularity and ability to represent interaction effects. While there is an emerging consensus that rule ensembles should optimize the trade-off between statistical risk and *cognitive complexity* in terms of number and lengths of rules (see Fig. 1), there is a multitude of diverse approaches for performing this optimization. This ranges from computationally inexpensive generate-and-select approaches [10; 14], over more expensive minimum-description length and Bayesian approaches [29], to expensive full-fledged discrete optimization methods [6; 30]. Within this range of options, methods based on *gradient boosting* [9] are of special interest because of their robustness against changes in the training data, flexibility to adapt to various response variable types and loss functions, and finally their good model performance relative to their computational cost.

On the other hand, state-of-the-art rule boosting approaches are based on design choices that compromise their risk/complexity trade-off. The traditional gradient boosting adaption [8] resorts to greedy optimization of the individual rules, which results in additional rules and additional conditions per rule to reach a desired statistical risk level. The more recent optimal rule boosting approach [3] partially addresses this issue, but it is based on the uncorrected weight updates of the extreme gradient boosting framework [4]. This too results in the inclusion of unnecessary extra rules, especially for

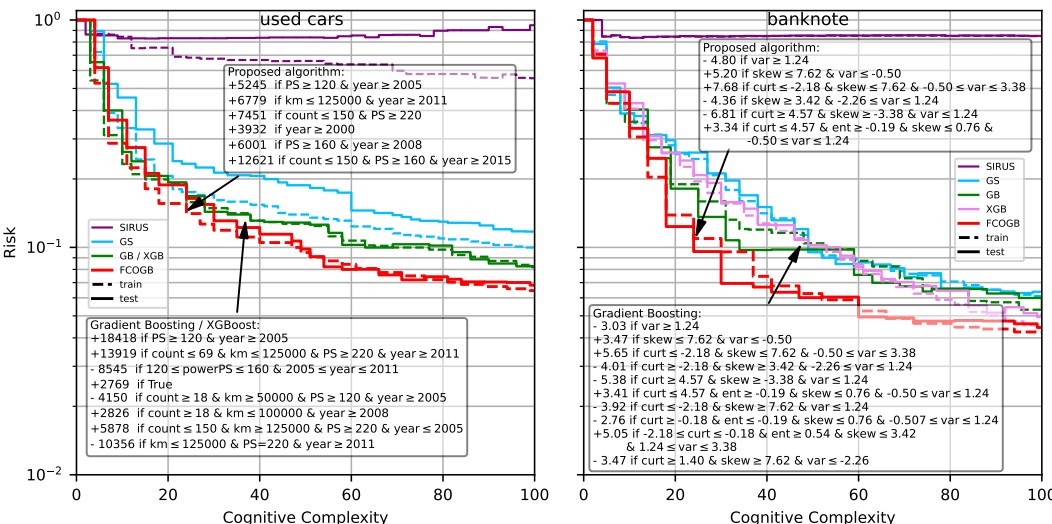

Figure 1: Risk/complexity curves for previous rule boosting variants (green) and proposed orthog­onalization approach (red) for dataset `used_cars` and `banknote`. The two highlighted corners correspond to rule ensembles with roughly equivalent training risk but substantially reduced cognitive complexity for the proposed algorithm.

loss functions with unbounded second derivatives like the Poisson loss. Most importantly, both approaches use the strict stagewise fitting approach where rules are not revised after they are added to the ensemble. Thus, they can increase accuracy only by adding further rules, even when the same gains can be achieved, in a more interpretable form, by altering those already present in the model.

Here we develop the first rule boosting algorithm that consistently optimizes the accuracy/complexity trade-off of the produced rule sets. For that, we adopt the fully corrective boosting approach [26] where all rule consequents are re-optimized in every boosting round, which can be done with only little computational extra effort given the usual convex loss functions. We then derive a new objective function for selecting individual rule bodies that anticipates the subsequent consequent re-optimization. This function is based on considering only the part of a rule body orthogonal to the already selected rules, which, as we show, correctly identifies the best approximation to the ideal space for consequent optimization defined by the risk gradient. Finally, we derive a corresponding efficient algorithm for cut-point search, which is crucial for, either greedy or branch-and-bound, single rule optimization. As we demonstrate on a wide range of datasets, the resulting rule boosting algorithm significantly outperforms the previous boosting variants in terms of risk/complexity trade-off, which can be attributed to a better risk reduction per rule as well as an affinity to select simpler rules. At the same time, the computational cost remains comparable to the previous branch-and-bound rule learner.

The paper is organized as follows. After giving a brief overview of the wider literature on intepretable machine learning and additive rule ensembles (Sec. 2), we recall the formal basics of rule ensembles and gradient boosting in Sec. 3. We then present our main technical contributions in Sec. 4 and their empirical evaluation in Sec. 5, before concluding in Sec. 6.

## 2 Related Literature

In contrast to post-hoc explanations of blackbox models [e.g., 28; 23], which are often unfaithful to the original model [21; 24; 13], interpretable machine learning methods aim to produce intrinsically intelligible, yet accurate, models. Additive models that compose terms in a simple summation are particularly useful in this context, because of their modularity, i.e., the possibility to comprehend the terms in isolation. As long as the individual terms are not too numerous and simulatable, i.e., their output can be approximately computed by a human, the resulting model is highly interpretable.

Good examples for this are (generalized) linear models [GLMs, 19], or generalized additive mod­els [GAMs, 12; 16]. However, they do not model variable interactions, at least not of higher order. Conjunctive propositional rules, on the other hand, have this ability, explaining their longstanding popularity in machine learning and related fields. Additive rule ensembles, which are closely related

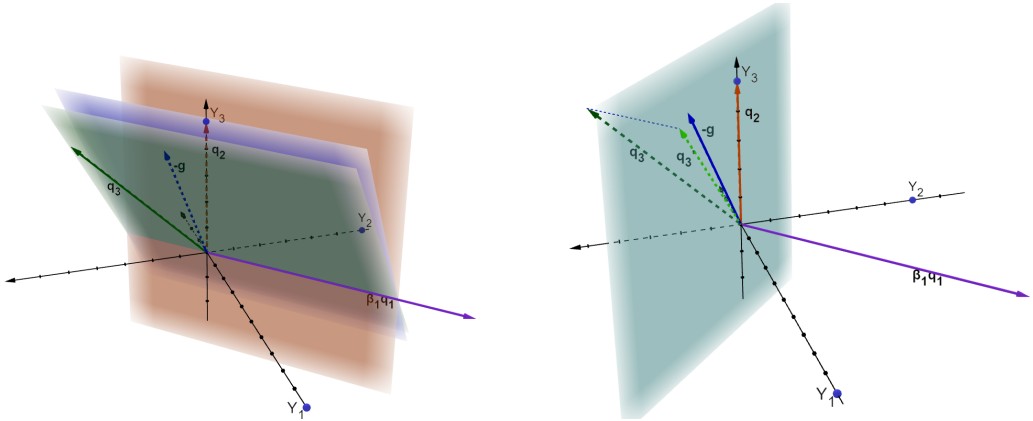

Figure 2: Illustration of output space for toy regression example with three data points with target values $y_1 = -10$, $y_2 = -6$, $y_3 = 5$ and three queries with output vectors $\mathbf{q}_1 = (1, 1, 0)$, i.e., $q_1$ selects the first two data points, $\mathbf{q}_2 = (0, 0, 1)$, and $\mathbf{q}_3 = (0, 1, 1)$. The gradient boosting objective selects $q_1$ with weight $\beta_1 = -8$ as first rule, resulting in a negative gradient vector $-\mathbf{g} = (-2, 2, 5)$. *Left:* Approximations (4) to target subspace (blue) spanned by $\mathbf{q}_1$ and $-\mathbf{g}$. The subspace (green) spanned by $\mathbf{q}_3$ and $\mathbf{q}_1$ is a better approximation than the subspace (orange) spanned by $\mathbf{q}_2$ and $\mathbf{q}_1$. However, the latter is selected by standard gradient boosting.*Right:* After projection onto orthogonal complement of already selected query, angle between $\mathbf{q}_3$ and $-\mathbf{g}$ is smaller than that between $\mathbf{q}_2$ and $-\mathbf{g}$ and is thus successfully selected by orthogonal gradient boosting objective.

to non-modular rule lists [e.g., 31; 22], thus provide a unique combination of interpretability and predictive power. There is a wide range of algorithms for learning additive rule ensembles. One approach is to generate a candidate set and then sub-selecting a rule ensemble, e.g. via sub-modular optimization [14; 32], or—as in RuleFit [10] or SIRUS [2]—via a sparse linear model, which is especially computationally inexpensive. However, these approaches are typically highly sensitive to the randomness in the generation of the candidate set. Alternatively, finding an optimal rule ensemble can be expressed as an integer program. Its relaxation as linear problem can then be solved via the column generation framework [6; 30], making the problem tractable. This approach is robust and flexible, but the full optimization remains computationally expensive.

Early approaches to additive rule ensemble learning that avoid initial candidate generation are based on the separate-and-conquer framework [11] and later on the original boosting algorithm [5; 17]. However, the first typically leads to non-modular rule lists and the second are designed for specific learning tasks only, typically classification. This problem is overcome with the gradient boosting framework [9], which generalizes the original AdaBoost algorithm [25] and allows fitting arbitrary differentiable loss functions. With this framework, rules are fitted stagewise based on their effect on the training loss when added to the ensemble [7; 8]. Extreme gradient boosting [4] increases the scalability of gradient boosting by avoiding numerical weight optimization. It is applicable whenever the loss function is twice differentiable. Fully-corrective boosting recalculate the weight of all weak learners after adding one weak learner into the ensemble model [26; 27]. It overcomes the drawback of the original gradient boosting algorithm that the weak learners are not changed after being generated. However, it is a high-level framework and does not solve the problem of how to select individual base learners.

## 3 Rule Boosting

An **additive ensemble** of $k$ rules can be represented by Boolean query functions $q_1, \ldots, q_k$ and a **weight vector** $\boldsymbol{\beta} = (\beta_1, \ldots, \beta_k)^T \in \mathbb{R}^k$ that jointly describe a function $f(\mathbf{x}) = \sum_{i=1}^{k} \beta_i q_i(\mathbf{x})$, the output of which can be mapped to the conditional distribution of a target variable $Y | X = \mathbf{x}$. That is, the queries define the rule antecedents (rule bodies), and the coefficients $\boldsymbol{\beta}$ define the rule consequents, i.e., the output of rule $i$ for input $\mathbf{x} \in \mathbb{R}^d$ is $\beta_i$ if $\mathbf{x}$ satisfies the antecedent, i.e., $q_i(\mathbf{x}) = 1$ (and 0 otherwise). Moreover, each **query function** $q_i : \mathbb{R}^d \to \{0, 1\}$ is a conjunction of $c_i$ **propositions**, i.e., $q_i(\mathbf{x}) = p_{i,1}(\mathbf{x}) p_{i,2}(\mathbf{x}) \ldots p_{i,c_i}(\mathbf{x})$ where the $p_{i,j}$ are typically a threshold function on an individual

99  input variable, e.g., $p_{i,j}(\mathbf{x}) = \delta(x_l \geq t)$. We denote the set of available propositions by $\mathcal{P}$ and the
100  **query language** of all conjunctions that can be formed from $\mathcal{P}$ as $\mathcal{Q}$.

101  We are concerned with two properties of an additive rule ensemble: its (empirical) prediction **risk**[1]
102  $R(f) = \frac{1}{n}\sum_{i=1}^{n} l(f(\mathbf{x}_i), y_i)$, measured by some positive loss function $l(f(\mathbf{x}), y)$ averaged over a
103  dataset $\{(\mathbf{x}_1, y_1), \ldots, (\mathbf{x}_n, y_n)\}$, and its **cognitive complexity** $C(f) = k + \sum_{i=1}^{k} c_i$, measuring
104  the cognitive effort required to parse all rule consequents and antecedents. Here we consider loss
105  functions that can be derived as negative log likelihood (or rather deviance function) when interpreting
106  the rule ensemble output as natural parameter of an exponential family model of the target variable,
107  which guarantees that the loss function is strictly convex and twice differentiable. Specifically, we
108  consider the cases of **squared loss** $l_{\mathrm{sqr}}(f(x_i), y_i) = (f(x_i) - y_i)^2$, the **logistic loss** $l_{\mathrm{log}}(f(x_i), y_i) =$
109  $\log(1 + \exp(-y_i f(x_i)))$, and the **Poisson loss** $l_{\mathrm{poi}}(f(x_i), y_i) = \log y_i - f(x_i) - y_i + \exp(f(x_i))$.

110  **Gradient boosting**   Gradient boosting [9] is a "stagewise" fitting scheme for additive models that,
111  in our context, produces a sequence of rule ensembles $f^{(0)}, f^{(1)}, \ldots, f^{(k)}$ such that $f^{(0)}(\mathbf{x}) = 0$ and,
112  for $t \in [1, k]$, $f^{(t)}(\mathbf{x}) = f^{(t-1)}(\mathbf{x}) + \beta_t q_t(\mathbf{x})$. Specifically, the term $\beta_t q_t(\mathbf{x})$ is chosen to perform an
113  approximate gradient descent with respect to the risk function $R(f) = R(\mathbf{f})$ considered as a function
114  of the model **output vector** $\mathbf{f} = (f(\mathbf{x}_1), \ldots, f(\mathbf{x}_n))$. The exact gradient descent update would be
115  $-\alpha_* \mathbf{g}$ where $\mathbf{g}$ is the **gradient vector** with components $g_i = \partial l(f(\mathbf{x}_i), y_i)/\partial f(\mathbf{x}_i)$ and $\alpha_*$ is the
116  step length that minimizes the empical risk $R(\mathbf{f} - \alpha \mathbf{g})$. However, since in general there is no query
117  $q$ for which the output vector $\mathbf{q} = (q(x_1), \ldots, q(x_n))$ is equal to the gradient $\mathbf{g}$, the goal is to select
118  $\mathbf{q}^*$ that best approximates $\mathbf{g}$ in the sense that it minimizes the squared **projection error**

$$\min_{\beta \in \mathbb{R}} \| -\alpha_* \mathbf{g} - \beta \mathbf{q} \|^2 = \alpha_*^2 \left( \|\mathbf{g}\|^2 - \frac{(\mathbf{q}^T \mathbf{g})^2}{\|\mathbf{q}\|^2} \right) \quad . \tag{1}$$

119  This is achieved by choosing $q_t$ to maximize the standard **gradient boosting objective** [8] $\mathrm{obj}_{\mathrm{gb}}(q) =$
120  $|\mathbf{q}^T \mathbf{g}|/\|\mathbf{q}\|$ and to find $\beta_t = \arg\min_{\beta \in \mathbb{R}} R(\mathbf{f} + \beta \mathbf{q}_t)$ via a line search. Note that this $\beta_t$ is not
121  generally equal to the minimizing $\beta$ in (1), because the optimal update in direction $\mathbf{q}$ can be better
122  than the best geometric approximation to the gradient descent update in direction $\mathbf{q}$. A derivation of
123  this objective function is the **gradient sum objective** [8; 26] $\mathrm{obj}_{\mathrm{gs}}(q) = |\mathbf{q}^T \mathbf{g}|$, which always selects
124  more general rules than the gradient boosting objective [8, Thm. 1], however, typically at the expense
125  of an increased risk per rule, because the correction of data points with large gradient elements has to
126  be toned down to avoid over-correction of other selected data points with small gradient elements.
127  Finally, an adaption of "extreme gradient boosting" [4] to rule ensembles yields the **extreme boosting**
128  **objective** [3] $\mathrm{obj}_{\mathrm{xgb}}(q) = (\mathbf{q}^T \mathbf{g})^2/\mathbf{q}^T \mathbf{h}$ where $\mathbf{h} = \mathrm{diag}(\nabla_{f(\mathbf{x})}^2 R(f))$ is the diagonal vector of the
129  risk Hessian again with respect to the output vector $\mathbf{f}$. This approach starts from the second order
130  approximation of $R(\mathbf{f} + \beta \mathbf{q})$ for which also yields a closed form weight update $\beta_t = -\mathbf{q}^T \mathbf{g}/\mathbf{q}^T \mathbf{h}$.
131  This approach is well-defined for our loss functions derived from exponential family response models,
132  which guarantee defined and positive $\mathbf{h}$. For the squared loss, it is equivalent to standard gradient
133  boosting, because the second order approximation is exact for $l_{\mathrm{sqr}}$ and $\mathbf{h}$ is constant.

134  **Single rule optimization**   While the rule optimization literature can be neatly divided into heuristic
135  (greedy / beam search) and exact branch-and-bound search, these approaches are actually closely
136  related: they can both be described as traversing a lattice on the query language $\mathcal{Q}$ imposed by a
137  **specialization relation** $q \preceq q'$ that holds if the propositions in $q'$ are a superset of those mentioned in
138  $q$, and $q'$ thus logically implies $q$. The difference between the approaches is under what conditions
139  they discard specializations of candidate queries and how those specializations are generated.

140  Here, we build on the branch-and-bound framework presented in Boley et al. [3] that allows to
141  efficiently search for optimal conjunctive queries in a condensed search space, given that there is
142  an admissible, effective, and efficiently computable bounding function for the employed objective.
143  Specifically, let $\mathrm{obj}\colon \mathcal{Q} \to \mathbb{R}$ denote the objective function to be maximized. Then a **bounding**
144  **function** $\mathrm{bnd}\colon \mathcal{Q} \to \mathbb{R}$ is **admissible** if for all $q \in \mathcal{Q}$ it holds that $\mathrm{bnd}(q) \geq \mathrm{bst}(q)$ where
145  $\mathrm{bst}(q) = \max_{q \preceq q' \in \mathcal{Q}} \mathrm{obj}(q')$ denotes the objective value of the best specialization of $q$. A bounding
146  function is effective in allowing to prune the search space if the difference $\mathrm{bnd}(q) - \mathrm{bst}(q)$ tends

---

[1]Note that for all algorithms discussed here, the $l_2$-regularized risk can also be considered with only light modifications. However, for ease of exposition and since regularization typically is not crucial for the rather small rule ensembles considered here, we focus on the unregularized case.

147 to be small, rendering $\mathrm{bst}(q)$ itself the theoretically most effective bounding function. However,
148 computing $\mathrm{bst}(q)$ is as hard as the overall optimization problem.

149 A frequently applied recipe for constructing an admissible bounding function that is also effective and
150 efficiently computable is to relax the quantifier in the definition of $\mathrm{bst}$ and instead of bounding the
151 value of the best specialization in the search space, bound the value of the best subset of data points
152 of those selected by $q$ [20]. This results in the tight bounding function when **unaware of selectability**

$$\mathrm{bnd}(q) = \max\{\mathrm{obj}(\mathbf{q}') : \mathbf{q}' \leq \mathbf{q}, \mathbf{q}' \in \{0,1\}^n\} \geq \mathrm{bst}(q) \ . \tag{2}$$

153 Here, $\mathbf{q} \leq \mathbf{q}'$ refers to the component-wise less or equals relation on the binary output vector of $q$
154 and $q'$. This function can be efficiently computed for many objective functions by pre-sorting the
155 data in time $O(n \log n)$ that has to be carried out only once per fitted rule [15]. For instance for the
156 extreme boosting objective, the optimum $q' \in \{0,1\}^n$ can be found as a prefix or suffix of all data
157 points after sorting them according to the ratio $g_i/h_i$ of first and second order loss derivatives.

# 4 Fully-corrective Orthogonal Gradient Boosting

159 Having reviewed the existing rule boosting approaches, we now turn to improving them in terms of
160 their risk/complexity trade-off. Our approach to this is to improve the risk reduction per rule added to
161 the ensemble, which directly affects the number of rules needed to achieve a certain risk. As it turns
162 out, this typically coincides with preferring the addition of more general and hence simpler rules.
163 Thus, it also positively affects the cognitive complexity on the level of the lengths of individual rules.

## 4.1 Weight Correction and Subspace Approximations

165 A natural idea to reduce the ensemble risk per rule added is to relax the strict stagewise fitting
166 approach of traditional gradient boosting and to allow the whole weight vector $\boldsymbol{\beta}$ to be adjusted in
167 every round $t$, i.e., to set

$$\boldsymbol{\beta}^{(t)} = \underset{\beta \in \mathbb{R}^t}{\arg\min}\, R(\mathbf{Q}_t \boldsymbol{\beta}) \ , \tag{3}$$

168 where $\mathbf{Q}_t = [\mathbf{q}_1, \ldots, \mathbf{q}_t]$ is the $n \times t$ **query matrix** with the output vectors of all selected queries as
169 columns. In contrast to a full joint optimization of queries and weights, this intermediate solution
170 still retains the computational benefits of gradient boosting for small rule ensembles: Given that our
171 loss function $l$ and therefore the empirical risk $R$ are convex, optimizing the weights in round $t$ is a
172 convex optimization problem in $t$ variables, equivalent to fitting a small generalized linear model.
173 Using Newton-Raphson ("iterated least squares") type algorithms, the computational cost of this is
174 usually negligible compared to the more expensive query optimization step, especially when aiming
175 for optimal individual queries for their reduced cognitive complexity.

176 Re-optimizing the weights, which is sometimes referred to as **fully corrective boosting** [26], ef-
177 fectively turns boosting into a form of forward variable selection for linear models. However, in
178 contrast to conventional variable selection where all variables are given explicitly, we still have to
179 identify a good query $q_t$ in each boosting iteration, and it turns out that finding the appropriate query
180 is more complicated as in the case of single weight optimization characterized by (1). We still would
181 like to add the direction of steepest descent, i.e., the negative gradient $-\mathbf{g}$, to the subsequent risk
182 optimization step and approximate as closely as possible the outcome $[\mathbf{Q}_{t-1}; \mathbf{g}]\boldsymbol{\alpha}^*$ where $\boldsymbol{\alpha}^* \in \mathbb{R}^t$
183 are the risk minimizing weights for $\mathbf{q}_1, \ldots, \mathbf{q}_{t-1}, \mathbf{g}$. Therefore, the best approximating query $q$ is
184 now given by

$$\underset{q \in \mathcal{Q}}{\arg\min}\, \underset{\beta \in \mathbb{R}^t}{\min}\, \|[\mathbf{Q}_{t-1}; \mathbf{g}]\boldsymbol{\alpha}^* - [\mathbf{Q}_{t-1}; \mathbf{q}]\boldsymbol{\beta}\|^2 \ . \tag{4}$$

185 It is an important observation that the standard gradient boosting objective does not correctly identify
186 this optimally approximating query. This is demonstrated by the example illustrated in Fig. 2. In
187 the left sub-figure it can be seen that the green plane, $\mathrm{span}\{\mathbf{q}_1, \mathbf{q}_3\}$, is a better approximation to
188 $\mathrm{span}\{\mathbf{q}_1, -\mathbf{g}\}$ (blue) than the orange plain, $\mathrm{span}\{\mathbf{q}_1, \mathbf{q}_2\}$. However, the latter is selected by standard
189 gradient boosting, because the angle between $\mathbf{q}_2$ and $-\mathbf{g}$ is smaller than that between $\mathbf{q}_3$ and $-\mathbf{g}$.

## 4.2 An Objective Function to Identify the Best Approximating Subspace

191 The intuitive reason for the gradient boosting objective failing to identify the correct query in Fig. 2
192 is that selecting $x_2$ in addition to $x_3$ is not beneficial for the overall risk reduction *if* we are only

allowed to set the weight for the newly selected query. This is because then this weight has to be a compromise between the two different magnitudes of correction required for $x_2$, which only needs a small positive correction, and $x_3$, which needs a large positive correction. If we, however, are allowed to change the weight of the previously selected query this consideration changes, because we can now balance an over-correction for $x_2$ by adjusting the weight of the first rule. While on first glance it seems unclear how much of such re-balancing can be applied without harming the overall risk, it turns out that this is captured by a simple criterion based on the norm of the part of the newly selected query that is orthogonal to the already selected ones.

**Lemma 4.1.** For $\mathbf{g} \in \mathbb{R}^n$, $\mathbf{Q} = [\mathbf{q}_1, \ldots, \mathbf{q}_{t-1}] \in \mathbb{R}^{n \times (t-1)}$, and $\mathbf{f} \in \mathrm{span}\{\mathbf{q}_1, \ldots, \mathbf{q}_{t-1}, \mathbf{g}\}$, we have

$$\underset{q \in \mathcal{Q}}{\arg\min} \min_{\boldsymbol{\beta} \in \mathbb{R}^t} \|\mathbf{f} - [\mathbf{q}_1, \ldots, \mathbf{q}_{t-1}, \mathbf{q}]\boldsymbol{\beta}\|^2 = \underset{q \in \mathcal{Q}}{\arg\max} \frac{|\mathbf{g}_\perp^T \mathbf{q}|}{\|\mathbf{q}_\perp\|} \ . \tag{5}$$

where for a vector $\mathbf{v} \in \mathbb{R}^n$ we denote by $\mathbf{v}_\perp$ its projection onto the orthogonal complement of range $\mathbf{Q}$. (All proofs of lemmas and theorems are in SI [1])

From this result we can directly derive $|\mathbf{g}_\perp^T \mathbf{q}|/\|\mathbf{q}_\perp\|$ as suitable objective function for fully corrective gradient boosting. However, it is worth incorporating two further observations. Firstly, we can show that, after applying the weight correction step (3), the gradient vector satisfies $\mathbf{g} = \mathbf{g}_\perp$, i.e., it is orthogonal to the subspace spanned by the selected queries, and therefore can be used in the objective function without projection.

**Lemma 4.2.** Let $\mathbf{g}$ be the gradient vector after the application of the weight correction step (3) for selected queries $\mathbf{q}_1, \ldots, \mathbf{q}_t$. Then $\mathbf{g} \perp \mathrm{span}\{\mathbf{q}_1, \ldots, \mathbf{q}_t\}$.

Moreover, the right hand side of Eq. (4) is technically undefined for redundant query vectors $\mathbf{q}$ that lie in range $\mathbf{Q}$ and therefore have $\|\mathbf{q}_\perp\| = 0$. Through Lm. 4.2 we know that for such queries we also have $\mathbf{g}^T \mathbf{q} = 0$, which suggests to simply fix this issue by defining the objective value in this case to be $0$. However, this solution would not fix numerical instabilities when $\|\mathbf{q}_\perp\|$ is close to zero. A better solution is therefore to add a small positive value $\epsilon$ to the denominator, which can be considered a **regularization parameter**. With this we arrive at the final form of our objective function, which we state along with some of its basic properties in the following theorem.

**Theorem 4.3.** *Let* $\mathbf{Q} = [\mathbf{q}_1, \ldots, \mathbf{q}_{t-1}] \in \mathbb{R}^{n \times (t-1)}$ *be the selected query matrix and* $\mathbf{g}$ *the corresponding gradient vector after full weight correction, and let us denote by* $\mathbf{q} = \mathbf{q}_\perp + \mathbf{q}_\|$ *the orthogonal decomposition of* $\mathbf{q}$ *with respect to* range $\mathbf{Q}$. *Then we have for a maximizer* $\mathbf{q}^*$ *of the* **orthogonal gradient boosting objective** $\mathrm{obj}_{\mathrm{ogb}}(q) = |\mathbf{g}^T \mathbf{q}|/(\|\mathbf{q}_\perp\| + \epsilon)$:

    *a) For* $\epsilon \to 0$, $\mathrm{span}\{\mathbf{q}_1, \ldots, \mathbf{q}_{t-1}, \mathbf{q}^*\}$ *is the best approximation to* $\mathrm{span}\{\mathbf{q}_1, \ldots, \mathbf{q}_{t-1}, \mathbf{g}\}$.

    *b) For* $\epsilon \to \infty$, $\mathbf{q}^*$ *maximizes* $\mathrm{obj}_{\mathrm{gs}}$ *and any maximizer of* $\mathrm{obj}_{\mathrm{gs}}$ *maximizes* $\mathrm{obj}_{\mathrm{ogb}}$.

    *c) For* $\epsilon = 0$ *and* $\|\mathbf{q}_\perp\| > 0$, *the ratio* $(\mathrm{obj}_{\mathrm{ogb}}(q)/\mathrm{obj}_{\mathrm{gb}}(q))^2$ *is equal to* $1 + (\|\mathbf{q}_\||/\|\mathbf{q}_\perp\|)^2$.

    *d) The objective value* $\mathrm{obj}_{\mathrm{ogb}}(q)$ *is upper bounded by* $\|\mathbf{g}\|$.

Intuitively, the orthogonal gradient boosting objective function measures the cosine of the angle between the gradient vector and the orthogonal projection of a candidate query vector $\mathbf{q}$. This is in contrast to the standard gradient boosting objective, which considers the angle of the unprojected query vector instead. In the example in Fig. 2 we can observe that this difference leads to successfully identifying the best approximating subspace, and Thm. 4.3a) guarantees this property.

## 4.3 Efficient Implementation

To develop an efficient optimization algorithm for the orthogonal gradient boosting objective, we recall that projections $\mathbf{q}_\perp$ on the orthogonal complement of range $\mathbf{Q}$ can be naively computed via $\mathbf{q}_\perp = \mathbf{Q}((\mathbf{Q}^T \mathbf{Q})^{-1}(\mathbf{Q}^T \mathbf{q}))$ where we placed the parentheses to emphasize that only matrix-vector products are involved in the computation—at least once the inverse of the Gram matrix $\mathbf{Q}^T \mathbf{Q}$ is computed. This approach allows to compute projections, and thus objective values, in time $O(nt + t^2)$ per candidate query after an initial preprocessing per boosting round of cost $O(t^2 n + t^3)$.

In a first step, this naive approach can be improved by maintaining an orthonormal basis of the range of the query matrix throughout the boosting rounds, resulting in a Gram-Schmidt-type procedure.

**Algorithm 1** Fully-corrective Orthogonal Gradient Boosting

**Input:** dataset $(x_i, y_i)_{i=1}^n$, desired number of rules $k$
Initialise $f^{(0)} = 0$
**for** $t = 1$ **to** $k$ **do**
$\quad \mathbf{g} = \left( \frac{\partial l(f^{(t-1)}(x_1), y_1)}{\partial f^{(t-1)}(x_1)}, \dots, \frac{\partial l(f^{(t-1)}(x_n), y_n)}{\partial f^{(t-1)}(x_n)} \right)$
$\quad q_t = \arg\max_{q \in \mathcal{Q}} \frac{|\mathbf{q}^T \mathbf{g}|}{\|\mathbf{q}_\perp\|}$ via $\texttt{beam}(\mathbf{g}, \mathbf{O}_{t-1})$ or $\texttt{bb}(\mathbf{g}, \mathbf{O}_{t-1})$
$\quad \mathbf{o}_t = \mathbf{q}_{t\perp} / \|\mathbf{q}_{t\perp}\|$ and $\mathbf{O}_t = [\mathbf{O}_{t-1}; \mathbf{o}_t]$
$\quad \boldsymbol{\beta}_t = \arg\min_{\boldsymbol{\beta} \in \mathbb{R}^t} R([\mathbf{q}_1, \dots, \mathbf{q}_t] \boldsymbol{\beta})$ via $\texttt{convex\_opt}$
$\quad f^{(t)} = [\mathbf{q}_1, \dots, \mathbf{q}_t] \boldsymbol{\beta}_t$
**Output:** $f^{(k)}(\cdot) = \beta_{k,1} q_1(\cdot) + \dots + \beta_{k,k} q_k(\cdot)$

Table 1: Comparison of normalised training risks and computation times for rule ensembles, averaged over cognitive complexities between 1 and 50, using SIRUS(SRS), Gradient Sum(GS), Gradient boosting (GB), XGBoost (XGB) and FCOGB, for benchmark datasets of classification (upper), regression (middle) and Poisson regression problems (lower).

| DATASET | d | n | TRAINING RISKS | | | | | TESTING RISKS | | | | | COMPUTATION TIMES | | | | |
|---|---|---|---|---|---|---|---|---|---|---|---|---|---|---|---|---|---|
| | | | SRS | GS | GB | XGB | FCOGB | SRS | GS | GB | XGB | FCOGB | SRS | GS | GB | XGB | FCOGB |
| TITANIC | 7 | 1043 | .895 | .662 | .635 | .637 | **.610** | .894 | .723 | .712 | .721 | **.707** | 7.077 | 2.624 | 9.858 | 10.21 | 25.71 |
| TIC-TAC-TOE | 27 | 958 | .892 | .741 | .627 | .640 | **.587** | .885 | .800 | .722 | .689 | **.669** | 12.59 | 3.971 | 10.34 | 6.09 | 13.99 |
| IRIS | 4 | 150 | .685 | .253 | .222 | .287 | **.218** | .745 | **.384** | .429 | .408 | .511 | 11.02 | 0.775 | 1.099 | 1.453 | 2.487 |
| BREAST | 30 | 569 | .569 | **.273** | .291 | .314 | .292 | .627 | **.273** | .370 | .376 | .348 | 11.48 | 6.744 | 74.43 | 74.83 | 239.2 |
| WINE | 13 | 178 | .578 | .162 | .216 | .192 | **.146** | .621 | .340 | .471 | .402 | **.242** | 9.456 | 1.530 | 4.432 | 2.154 | 55.18 |
| IBM HR | 32 | 1470 | .980 | .572 | .560 | .573 | **.560** | .974 | .607 | .618 | .626 | **.606** | 11.15 | 17.24 | 10.99 | 12.92 | 12.03 |
| TELCO CHURN | 18 | 7043 | .944 | .679 | .683 | .679 | **.670** | .945 | .663 | .677 | .673 | **.663** | 50.83 | 40.01 | 1883 | 1485 | 3039 |
| GENDER | 20 | 3168 | **.566** | .996 | .996 | .996 | .996 | **.570** | 1.000 | 1.000 | 1.000 | 1.000 | 22.42 | 22.73 | 25.49 | 24.27 | 32.95 |
| BANKNOTE | 4 | 1372 | .854 | .303 | .264 | .288 | **.227** | .858 | .310 | .263 | .297 | **.228** | 8.933 | 6.298 | 5.648 | 7.060 | 8.444 |
| LIVER | 6 | 345 | .908 | .809 | .800 | .787 | **.777** | .917 | **.913** | 1.000 | .928 | 1.000 | 9.734 | 1.997 | 99.72 | 124.1 | 193.9 |
| MAGIC | 10 | 19020 | .906 | .720 | .709 | .710 | **.707** | .903 | .702 | .693 | .693 | **.687** | 1.364 | 75.14 | 89.18 | 101.9 | 352.2 |
| ADULT | 11 | 30162 | .804 | .594 | .599 | .594 | **.582** | .802 | .603 | .615 | .607 | **.597** | 2.169 | 121.0 | 136.7 | 146.0 | 728.3 |
| DIGITS5 | 64 | 3915 | **.248** | .331 | .312 | .335 | .353 | **.262** | .329 | .314 | .330 | .350 | 52.60 | 110.8 | 72.74 | 101.5 | 97.4 |
| INSURANCE | 6 | 1338 | .169 | .144 | .143 | .146 | **.126** | .177 | .134 | .137 | .140 | **.126** | 14.06 | 7.507 | 15.94 | 12.98 | 39.53 |
| FRIEDMAN1 | 10 | 2000 | .180 | .069 | .073 | .071 | **.068** | .165 | **.072** | .080 | .079 | .074 | 16.79 | 2.514 | 4.302 | 3.171 | 6.915 |
| FRIEDMAN2 | 4 | 10000 | **.082** | .092 | .119 | .116 | .101 | **.082** | .094 | .120 | .117 | .101 | 47.33 | 11.79 | 17.56 | 13.18 | 28.4 |
| FRIEDMAN3 | 4 | 5000 | .093 | .044 | .043 | .043 | **.041** | .092 | .046 | .046 | .046 | **.044** | 29.86 | 6.243 | 10.61 | 8.559 | 17.65 |
| WAGE | 5 | 1379 | .427 | .368 | .366 | .355 | **.342** | **.341** | .377 | .397 | .394 | .396 | 14.18 | 5.605 | 12.12 | 13.17 | 25.19 |
| DEMOGRAPHICS | 13 | 6876 | .219 | .214 | .213 | .213 | **.212** | **.209** | .217 | .217 | .217 | .216 | 38.24 | 36.80 | 29.40 | 33.04 | 72.42 |
| GDP | 1 | 35 | .063 | .024 | .024 | .024 | **.024** | .059 | .025 | .025 | .025 | **.025** | 7.974 | .261 | .351 | .282 | .488 |
| USED CARS | 4 | 1770 | **.175** | .266 | .250 | .251 | .225 | **.172** | .289 | .265 | .271 | .241 | 15.00 | 8.371 | 12.10 | 9.484 | 20.27 |
| DIABETES | 10 | 442 | .156 | .137 | .137 | .136 | **.130** | .188 | **.148** | .150 | .155 | .158 | 10.50 | 2.204 | 3.574 | 3.920 | 7.591 |
| BOSTON | 13 | 506 | .101 | .089 | .090 | .087 | **.081** | .105 | **.078** | .086 | .086 | .081 | 10.96 | 3.055 | 6.731 | 5.285 | 10.44 |
| HAPPINESS | 8 | 315 | .109 | .031 | .031 | .032 | **.030** | .109 | **.033** | .038 | .038 | .037 | 6.344 | 1.160 | 11.37 | 11.31 | 26.43 |
| LIFE EXPECT. | 21 | 1649 | .109 | .026 | .026 | .026 | **.025** | .110 | .027 | .027 | .027 | **.026** | 21.44 | 16.16 | 58.43 | 63.82 | 131.2 |
| MOBILE PRICES | 20 | 2000 | .148 | **.131** | .137 | .137 | .135 | .140 | .136 | .143 | .145 | .142 | 33.81 | 15.03 | 367.7 | 442.5 | 815.4 |
| SUICIDE RATE | 5 | 27820 | .547 | .543 | .540 | .541 | **.534** | **.514** | .521 | .521 | .521 | .515 | 52.35 | 109.6 | 117.1 | 139.6 | 644.6 |
| VIDEOGAME | 6 | 16327 | **.895** | .953 | .953 | .953 | .953 | .850 | .720 | .720 | .720 | **.720** | 1.171 | 41.91 | 34.38 | 45.90 | 119.1 |
| RED WINE | 11 | 1599 | .072 | .034 | .035 | .035 | **.034** | .073 | .035 | .035 | .036 | **.035** | 19.94 | 9.149 | 15.32 | 21.99 | 35.34 |
| COVID VIC | 4 | 85 | NA | .144 | .130 | .368 | **.125** | NA | .523 | .600 | .628 | **.130** | NA | .127 | .382 | | .854 |
| COVID | 2 | 225 | NA | **.341** | .374 | 2.893 | .347 | NA | **.447** | .482 | 4.115 | .469 | NA | .701 | .690 | .682 | 1.143 |
| BICYCLE | 4 | 122 | NA | .352 | .317 | .310 | **.300** | NA | **.413** | .439 | .440 | .467 | NA | .695 | 1.103 | 1.105 | 2.124 |
| SHIPS | 4 | 34 | NA | .155 | .168 | 87.31 | **.145** | NA | **.222** | .288 | 109.1 | .420 | NA | .235 | .296 | .311 | .448 |
| SMOKING | 2 | 36 | NA | .114 | .109 | .165 | **.090** | NA | **.130** | .193 | .258 | .169 | NA | .266 | .256 | .208 | .301 |

Since the projections $\mathbf{q}_{t\perp}$ of the selected queries naturally form an orthogonal basis of $\text{range}\,\mathbf{Q}$ this only requires normalization, which can be done essentially without additional cost. Formally, by storing $\mathbf{o}_t = \mathbf{q}_{t\perp} / \|\mathbf{q}_{t\perp}\|$ in all boosting rounds $t$, subsequent projections can be computed via $q_\perp = \mathbf{q} - \mathbf{O}(\mathbf{O}^T \mathbf{q})$ where $\mathbf{O} = [\mathbf{o}_1, \dots, \mathbf{o}_t]$. This reduces the computational complexity per candidate query to $O(tn)$ without requiring any additional preprocessing.

While this looks like the optimal complexity for evaluating $\text{obj}_{\text{ogb}}$ in isolation, it leads to a prohibitive complexity for large $n$ for finding the optimal query in a given round. Specifically, branch-and-bound with the tight bounding function (2) evaluates $O(n)$ queries per expanded search node and beam search even $O(d^2 n)$. In both cases, using the expression for $\mathbf{q}_\perp$ above repeatedly results in a quadratic cost in $n$ per search node. To circumvent this, we have to exploit the structure of candidate evaluations, similar to other efficient implementations of rule and tree learning algorithms [18].

The candidates evaluated per search node of both branch-and-bound and beam search have query vectors that are prefixes of an ordered sub-selection of data points, in beam search because optimum cut-off values are sought for each of the $d$ input variables, in branch-and-bound because the optimum in Eq. (2) is attained or approximated by a prefix with respect to some presorting order. Hence, we need to solve the following **prefix optimization problem**: given an ordered sub-selection of $l$ of the $n$ data points, represented by an injective map $\sigma\colon \{1,\ldots,l\} \to \{1,\ldots,n\}$, find the optimal prefix

$$i_* = \underset{i\in\{1,\ldots,l\}}{\arg\max} \frac{\mathbf{g}^T\mathbf{q}^{(i)}}{\|\mathbf{q}_\perp^{(i)}\| + \epsilon} \tag{6}$$

where $\mathbf{q}^{(0)} = \mathbf{0}$ and $\mathbf{q}^{(i)} = \mathbf{q}^{(i-1)} + \mathbf{e}_{\sigma(i)}$. The following proof shows how the computational complexity for solving (6) can be substantially reduced compared to the direct approach above. It uses an incremental computation of projections that works directly on the available orthonormal basis vectors $\mathbf{o}$ instead of computing matrix-vector products or, even worse, the whole projection matrix.

**Theorem 4.4.** *Given a gradient vector* $\mathbf{g} \in \mathbb{R}^n$*, an orthonormal basis* $\mathbf{o}_1,\ldots,\mathbf{o}_t \in \mathbb{R}^n$ *of the subspace spanned by the queries of the first* $t$ *rules, and a sub-selection of* $l$ *candidate points* $\sigma\colon [l] \to [n]$*, the best prefix selection problem* (6) *can be solved in time* $O(tn)$*.*

*Proof sketch.* We write the objective value of prefix $i$ in terms of incrementally computable quantities:

$$\frac{\mathbf{g}^T\mathbf{q}^{(i)}}{\|\mathbf{q}_\perp^{(i)}\| + \epsilon} = \frac{\mathbf{g}^T\mathbf{q}^{(i)}}{\|\mathbf{q}^{(i)}\| - \|\mathbf{q}_\|^{(i)}\| + \epsilon} = \frac{\mathbf{g}^T\mathbf{q}^{(i)}}{\|\mathbf{q}^{(i)}\| - \sqrt{\sum_{k=1}^t \|\mathbf{o}_k\mathbf{o}_k^T\mathbf{q}^{(i)}\|^2} + \epsilon}\ \ .$$

In particular, the $t$ sequences of norms $\|\mathbf{o}_k\mathbf{o}_k^T\mathbf{q}^{(i)}\|$ can be computed in time $O(n)$ via cumulative summation of the $k$-th basis vector elements in the order given by $\sigma$:

$$\|\mathbf{o}_k\mathbf{o}_k^T\mathbf{q}^{(i)}\| = \|\mathbf{o}_k\| \sum_{j=1}^i \mathbf{o}_k^T\mathbf{e}_{\sigma(j)} = \sum_{j=1}^i o_{k,\sigma(j)}$$

$\square$

We close this section with a pseudocode (Alg. 1) that summarizes the main ideas of the orthogonal gradient boosting and refer to the literature for details about the beam-search/branch-and-bound step.

# 5  Experiments

In this section, we present empirical results comparing the proposed fully corrective orthogonal gradient boosting (FCOGB) to the standard gradient boosting algorithms [8] using greedy optimization of $\mathrm{obj}_{\mathrm{gb}}$ (GB) and $\mathrm{obj}_{\mathrm{gs}}$ (GS), to extreme gradient booosting [3] using branch-and-bound optimisation of $\mathrm{obj}_{\mathrm{xgb}}$ (XGB), and finally to SIRUS [2] as the state-of-the-art generate-and-filter approach. We investigate the risk/complexity trade-off, the affinity to select general rules, as well as the computational complexity. The datasets used are those of Boley et al. [3] augmented by three additional classification datasets from the UCI machine learning repository and, to introduce a novel modelling task to the rule learning literature, five counting regression datasets from public sources. This results in a total of 34 datasets (13 for classification, 16 for regression, and 5 for counting/Poisson regression, see Tab. 1). All algorithms were run five times on all datasets using 5 random 80/20 train/test splits to calculate robust estimates of all considered metrics. In all cases, the number of gradient boosting iterations was chosen to produce ensembles with cognitive complexity of at most 50. The experiment code and further information about the datasets are available on GitHub[1].

**Cognitive complexity versus risk**   Tab. 1 compares the complexity/risk trade-off of the boosting variants and SIRUS by the normalized risk averaged across all considered cognitive complexity levels (where normalization is performed by the risk of the empty rule ensemble). FCOGB has the smallest training risk for 26 of the 34 datasets, occasionally outperforming the second best algorithm by a wide margin (*tic-tac-toe*, *wine*, *banknote*, *insurance*, *boston*, *ships*, *smoking*). For the test risk the picture is more ambiguous, however, FCOGB retains a relative majority of datasets won. Performing one-sided paired t-tests at significance level 0.05 (with Bonferroni correction for 8 hypotheses) reveals that FCOGB significantly outperforms all other variants with a margin of at least 0.001 average normalized training risk (while there is no significant winner in terms of test risk—likely due to a lack of regularization for larger ensemble sizes).

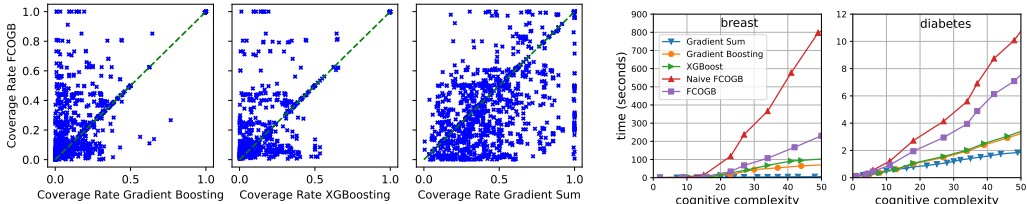

Figure 3: First three: the comparison of the coverage rate of Gradient Boosting, XGBoost, Gradient Sum and FCOGB. The upper (resp. lower) half of the green line means the coverage rate of FCOGB is higher (resp. lower) than the other method. Last two: the comparison of the running time of Gradient boosting, XGBoost and FCOGB for the benchmark datasets `breast cancer` and `diabetes` of generating rule ensembles with cognitive complexity 50.

**Coverage**  To compare the generality of the rules learned by the new objective function in comparison to the existing ones, we performed an additional experiment where we first used one of the previous objective functions to generate rule ensembles with ten rules for all datasets. Then for each partial rule ensemble, we applied the orthogonal gradient boosting objective function to find an alternative rule. Importantly, we used branch-and-bound with admissible bounding functions for all the alternative objectives to avoid confounding through sub-optimal greedy solutions. In Fig. 3 we compare the relative coverage, i.e., the relative number of selected data points $\|\mathbf{q}\|^2/n$, of the rules discovered by the original algorithms to the ones discovered by FCOGB. The outcome is that 81.1% of the FCOGB rules covers more data points than gradient boosting, and similarly 71.3% of its rules cover more data points than those generated by XGBoost. In contrast, only 47.2% of the FCOGB rules cover more datapoint than the ones discovered by gradient sum. These results are in alignment with the theoretical expectation in terms of the influence of the coverage on the objective values where gradient sum is completely unaffected, whereas orthogonal gradient boosting has a denominator that tends to grow with coverage albeit less than the one of gradient boosting.

**Computation time**  We also compare the computational cost of generating rule ensembles with cognitive complexity 50 by different algorithms in Tab 1. Comparing the computational cost of FCOGB to XGB, the other algorithm utilizing the more expensive branch-and-bound search, the costs are in the same order of magnitude except for one extreme case (wine) where FCOGB is a factor of 26 slower. Comparing to the two greedy variants, FCOGB is in the same order of magnitude as gradient boosting for most datasets. Unsuprisingly, there are a few examples where greedy search vastly outperforms branch-and-bound, in one case (telco churn) by a factor of around 76. However, overall, the results confirm that branch-and-bound search is a practical algorithm in absolute terms: For 23 benchmark dataset, FCOGB is able to finish training a model of cognitive complexity of 50 within one minute. Most of the other experiments run within 15 minutes except one dataset (telco churn) which require longer running time. Finally, Fig. 3 shows the detailed computation time of all algorithms in terms of cognitive complexity, including the naive implementation of FCOGB for *breast cancer* and *diabetes*, which shows that the performance improvement through Thm. 4.4 is critical to bring the computational complexity on par with XGB.

## 6 Conclusion

The proposed fully corrective orthogonal boosting approach is a worthwhile alternative to previously published boosting variants for rule learning, especially when targeting a beneficial risk-complexity trade-off and an overall small number of rules. The present work provided a relatively detailed theoretical analysis of the newly developed rule objective function. However, some interesting questions were left open. While the presorting-based approach to the bounding function performs extremely well in synthetic experiments, a theoretical approximation guarantee for this algorithm has yet to be derived. Another interesting direction for future work is the extension of the introduced approximating subspace paradigm to the extreme gradient boosting approach, which, due to the utilization of higher order information, should principally be able to produce even better risk-complexity trade-offs.

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
