# OpenReview forum: "Orthogonal Gradient Boosting for Interpretable Additive Rule Ensembles"
_NeurIPS.cc/2023/Conference — Submitted to NeurIPS 2023_

### Official Review · Reviewer_9SvB · 2023-07-02

**Soundness:** 3 good
**Presentation:** 2 fair
**Contribution:** 3 good
**Rating:** 5
**Confidence:** 4

**Summary:**

In this paper, the authors propose a gradient-boosting algorithmic framework for rule ensemble learning, emphasizing the interpretability of produced rule set. Various gradient-boosting algorithms are reviewed in the rule-learning context, and the authors argue that a specific boosting algorithm, called fully corrective orthogonal gradient boosting (FCOGB), is particularly suited for rule boosting. The intuition is that existing additive rule-boosting procedures operate in a strictly greedy fashion - the weight of each added rule is fixed in later iterations. In contrast, FCOGB allows the weights of preceding rules to be adjusted in each later iteration, which may help to reduce the number of required rules (to reach a certain accuracy) and thus the cognitive complexity of the final rule set. Based on FCOGB, the authors derive the stepwise boosting objective function for single rule search, which is similar to existing gradient boosting objectives but with a different regularization term. The overall algorithm looks like the conjugate gradient method - in each iteration, the rule aligning best with the gradient in the orthogonal complement of the subspace spanned by previous rules is added. The authors demonstrate the effectiveness of FCOGB through experimental comparison with existing rule-boosting algorithms on classification, regression, and Poisson regression tasks.

**Strengths:**

- Applying FCOGB to rule learning, to the best of my knowledge, is a novel idea, and the authors provide a comprehensible justification for this choice. Figure 2 is helpful in understanding the difference between FCOGB and existing rule-boosting algorithms.
- How to search the optimal rule in each iteration is especially considered, which is a key step in rule boosting. The authors propose a strategy that exploits the nice structure in the boosting objective function to speed up the bound calculation in branch-and-bound search of optimal rules.
- The proposed algorithm is evaluated on a wide range of datasets and tasks. The authors provide a detailed analysis of the results. Figure 1 clearly shows that FCOGB can achieve a better accuracy-risk trade-off than existing rule-boosting algorithms.


**Weaknesses:**

- The presentation of the paper can be improved. For example:
  + In the "Rule Boosting" section, the "Gradient boosting" subsection mixes the description of general gradient boosting and the more specific rule boosting. This makes it hard to understand these objectives for readers who are not familiar with the rule-boosting literature. For example, obj_gb(q) = |g^T q|/||q|| is nonstandard in the general gradient boosting literature. It would be better to separate the general gradient boosting and rule boosting parts.
  + The "Single rule optimization" subsection assumes too much prior knowledge about the rule learning literature. I would suggest the authors merge this subsection with the "4.3 Efficient Implementation" subsection to make the paper more fluent and self-contained.
  + The authors should provide more details about the proposed algorithm, especially the BnB/beam search of a single rule.
- Lack of comparison with rule induction algorithms based on column generation, e.g., [30] and [b]. In the column generation approach, the weights of all added rules are also adjusted in each iteration when solving the restricted master problem, which is similar to FCOGB. I am interested in how FCOGB compares with this approach experimentally.
- The presentation of the prefix optimization problem is misleading. The authors claim that "This function can be efficiently computed for many objective functions by pre-sorting the data in time O(n log n)" in Section 3, but is this true for the objective function obj_{ogb}(q)?  The authors should clarify this point. I cannot immediately see how the optimal solution to (2) under this objective function is contained in the prefix of the data sorted by some (what?) criterion. If this is true, the authors should provide a proof or a reference to support this claim.
- There is a mistake in Lines 233-234.
- Missing references:
  + [a] Jonathan Eckstein, Noam Goldberg. An Improved Branch-and-Bound Method for Maximum Monomial Agreement. INFORMS Journal on Computing, 2012.
  + [b] Jonathan Eckstein, Ai Kagawa, Noam Goldberg. REPR: Rule-Enhanced Penalized Regression. INFORMS Journal on Optimization, 2019.
  + [c] Fan Yang, et al. Learning Interpretable Decision Rule Sets: A Submodular Optimization Approach. NeurIPS 2021.

**Questions:**

- The test accuracy of FCOGB is not generally better than that of existing rule-boosting algorithms. Can this be explored in more detail?
- How the averaged, normalised training risks in Table 1 are computed? An exact formula would be helpful.
- It is well-known in recent rule learning literature ([6] and [c]) that the tic-tac-toe dataset can be perfectly learned by eight 3-CNF rules. Can FCOGB learn the perfect rule set on this dataset?

**Limitations:**

The limitations of this work are not explicitly discussed.

---

> ### Author Rebuttal · Authors · 2023-08-09
>
> We thank you for your thoughtful comments and generally positive evaluation. Below we first address your questions and then provide some clarifications regarding your other concerns.
>
> **Test performance**
>
> The results in the original submission unfortunately strongly undersold the proposed method in terms of test risk because they were based on the methods’ performance without regularization. This affected the proposed method the most, as, due to its improved objective, it fits the training data tightest and is therefore most prone to overfitting if unregularised. *Please have a look at the global rebuttal and the included pdf to see that, when appropriate L2-regularization is performed, the proposed method is the best in terms of test risk for 23 of the 34 datasets, and its advantage over the other boosting variants is statistically significant for both training and test risk (with appropriate Bonferroni correction)*. We hope that these additional results positively affect your overall evaluation of our work.
>
> NB The initial experiment was designed without regularization for the sake of a streamlined process and to focus on the effect of the new objective function in train error minimisation (which it directly affects as opposed to test risk, which is only affected indirectly).
>
> **Train / test risk formula**
>
> For each number of rules $k$, the normalized train and test risks are computed as the ratio of the risk of the ensemble $f_k$ to the base risk $f_0$ of the empty ensemble, i.e.,  $R(f_k)/R(f_0)$. This normalization was performed to make the results for different datasets comparable.
>
> **Learned rules for tic-tac-toe**
>
> This is indeed an interesting benchmark for the comparison of statistical to logical learners. Unfortunately, the proposed method does not learn the complete rule set of tic-tac-toe, neither within the first six rules nor within the first eleven rules. This can be explained by the overall greedy nature of the boosting approach combined with objective function that rewards picking relatively general "statistical" rules. For instance all boosting variants pick as first rule (weight varies based on regularization parameter):
>
> `+1.388 if middle-middle=x`
>
> Within the class of the considered statistical learner, we can see that the proposed algorithm does substantially improve the risk / complexity trade-off over the other boosting variants and SIRUS (see results).
>
> **Core contributions of the paper and greedy bound computation**
>
> We would like to clarify these two issues that we did not communicate clearly in the submitted version of the paper. After the derivation of a novel objective function that anticipates weight corrections, the second technical core contribution of the paper is to provide an algorithm that allows to efficiently compute objective values for incremental subsets of data points in a time that depends only linearly on the number of data points $n$ (in contrast to the quadratic dependency on $n$ that results from a naive implementation). Solving this problem is central for both, branch-and-bound search where typically the bounding function is approximated by optimizing over a prefix sequence of pre-sorted data points as well as greedy search where incremental cut-points have to be evaluated per input variable.
>
> We then opted to “package” the novel objective function with branch-and-bound search, because it can be expected that this choice emphasizes the differences with the previous objective functions better (in particular Ref 3, which also uses branch-and-bound). However, this branch-and-bound approach is, as of now, only empirical/heuristic in nature. While we did mention this issue in the conclusion section and in detail in the supplementary information, the main text itself was indeed misleading. We are eager to correct that in the published version and thank the reviewer for pointing this out.
>
> To provide more details, we use the prefix order with respect to the sums of selected gradient statistics divided by the norms of the corresponding query vectors. In extensive numerical experiments with subset sizes of up to 20 we find that the prefix greedy approach approximates the optimal subset objective value with at least a rate of 0.75 and in 90% of the cases with a rate of at least 0.9. These experiments are summarized in Section B of the supplementarity information (and details can be found in the notebook “analysis/greedy_analysis.ipynb” in the submitted codebase). Based on these results, we effectively use the branch-and-bound search as heuristic 0.75-factor approximation algorithm. As stated in the conclusion, we consider coming up with rigorous bounds as one of the most interesting questions related to the new objective function that are currently unanswered.
>
> **Comparison with column generation methods**
>
> We agree that the column generation approach is very interesting and are eager to compare its performance to boosting. Preliminary results based on seven datasets and our own ad hoc implementation of Ref 30 using Gurobi (we were not able to obtain an implementation from the authors) show the following: When capping the computation time to not exceed that of boosting too much (1000s for an individual fit), column generation does not improve the complexity / accuracy trade-off, in particular it tends to generate longer rules and to not or only slightly improve training risk. Given the intricacies of the implementation and the design of an insightful experimental setup (e.g., with reasonable caps on computation time) as well as the substantial amount of space required for a satisfactory exposition, we believe that this comparison deserves its own full paper with theoretical discussion and is out-of-scope of the current work, which primarily focusses on the comparison of boosting objectives.
>
> **Other**
>
> Thank you for pointing out the missing references and suggestions for improving the presentation. These will be incorporated.

---

> > ### Comment · Reviewer_9SvB · 2023-08-17
> >
> > Thanks for your detailed clarification and additional experiments. Given that the prefix greedy approach is heuristic and the 0.75-factor approximation is only justified by empirical observation, I will keep my score unchanged.

---

### Official Review · Reviewer_crnT · 2023-07-04

**Soundness:** 3 good
**Presentation:** 3 good
**Contribution:** 3 good
**Rating:** 6
**Confidence:** 3

**Summary:**

This paper introduces a novel approach to gradient boosting of decision rules for interpretable machine learning models. By incorporating a
weight correction step and orthogonal projections, the method maximizes predictive gain per rule.
Their experimental evaluation on various classification, regression, and Poisson regression tasks confirms that the resulting rule learner
enhances the trade-off between comprehensibility and accuracy in the fitted ensemble. Moreover, it maintains a comparable computational
cost to previous branch-and-bound rule learners.

**Strengths:**

1. Originality: The paper introduces the first rule boosting algorithm that consistently optimizes the accuracy/complexity trade-off of produced rule sets. This represents a novel contribution to the field.
2. Quality: The research exhibits high quality as it adopts the fully corrective boosting approach, which entails re-optimizing all rule consequents in each boosting round. The study's rigorous algorithm development provides a strong foundation for the research, ensuring
the reliability and robustness of the findings.
3. Clarity: The paper explains the new objective function for selecting individual rule bodies, the corresponding efficient algorithm for cutpoint
search along with some other algorithm details. The clear explanations contribute to the overall clarity of the research.
4. Significance: The research demonstrates significant improvements over previous boosting variants in terms of the risk/complexity tradeoff.
The better risk reduction per rule and the affinity to select simpler rules contribute to the overall significance of the findings. Additionally,
the comparable computational cost to previous approaches adds to the practical relevance of the research.

**Weaknesses:**

In terms of the compared established methods, SIRUS [1] is the most recent work included in the analysis, published in 2021. However, it is
worth noting that some more recent publications, such as [2,3], are not included in the experiment section.
One limitation of the paper's presentation is the heavy reliance on text and equations, with less emphasis on the use of figures and intuitive
example case studies. This approach may hinder the reader's ability to grasp complex concepts and visualize the practical applications of
the proposed methods. Incorporating more visual aids, such as figures and illustrative examples, could enhance the clarity and accessibility
of the research.
[1] C. Bénard, G. Biau, S. Da Veiga, and E. Scornet. Interpretable random forests via rule extraction. In International Conference on Artificial
Intelligence and Statistics, pages 937–945. PMLR, 2021.
[2] Souza V F, Cicalese F, Laber E, et al. Decision Trees with Short Explainable Rules[J]. Advances in Neural Information Processing
Systems, 2022, 35: 12365-12379.
[3] Calzavara S, Cazzaro L, Lucchese C, et al. Explainable Global Fairness Verification of Tree-Based Classifiers[C]//2023 IEEE Conference
on Secure and Trustworthy Machine Learning (SaTML). IEEE, 2023: 1-17.

**Questions:**

Could you provide some insights into the reason why certain newly published works, such as [1,2], were not included in the experiment
section? Were there any specific criteria or limitations that influenced the selection of methods for comparison?
[1] Souza V F, Cicalese F, Laber E, et al. Decision Trees with Short Explainable Rules[J]. Advances in Neural Information Processing
Systems, 2022, 35: 12365-12379.
[2] Calzavara S, Cazzaro L, Lucchese C, et al. Explainable Global Fairness Verification of Tree-Based Classifiers[C]//2023 IEEE Conference
on Secure and Trustworthy Machine Learning (SaTML). IEEE, 2023: 1-17.

**Limitations:**

A limitation of the study is that while it includes the most recent work, SIRUS [1], which was published in 2021, it does not incorporate some
more recent publications like [2,3] in the experiment section. This omission limits the comprehensiveness of the analysis and may overlook
potential advancements or alternative approaches introduced in these newer works.
[1] C. Bénard, G. Biau, S. Da Veiga, and E. Scornet. Interpretable random forests via rule extraction. In International Conference on Artificial
Intelligence and Statistics, pages 937–945. PMLR, 2021.
[2] Souza V F, Cicalese F, Laber E, et al. Decision Trees with Short Explainable Rules[J]. Advances in Neural Information Processing
Systems, 2022, 35: 12365-12379.
[3] Calzavara S, Cazzaro L, Lucchese C, et al. Explainable Global Fairness Verification of Tree-Based Classifiers[C]//2023 IEEE Conference
on Secure and Trustworthy Machine Learning (SaTML). IEEE, 2023: 1-17.

---

> ### Author Rebuttal · Authors · 2023-08-09
>
> We thank you for the positive evaluation of our work as well as for pointing out recent developments in the related literature.
>
> Based on a first assessment the work of Calzavara et al. is a related but clearly distinct problem: the extraction of a rule set from a single tree that is causally fair (wrt to some definition) but not necessarily optimizing the trade-off between rule set size and predictive performance. The other work of Souza et al. could be an interesting point of comparison in the future that is representative for the line of work of generating small individual decision trees. However, note that to our understanding this work is concerned only with classification trees that perfectly classify the given training data. Hence it does not directly fit into our empirical comparison, which, in contrast, considers methods that can learn arbitrary statistical response models as long as the target variable is modeled conditionally with an exponential family distribution. In particular, we use logistic regression, least squares regression, and Poisson regression, which can all not be treated by the approach described in Souza et al.
>
> Moreover, we appreciate the idea of including more illustrations in the work. The uploaded pdf in the general rebuttal shows more examples of actual rule ensembles and the effect of the new algorithm on the whole size/accuracy trade-off curve. We plan to include these either in the updated appendix or even in the main text if sufficient room can be made.

---

> > ### Comment · Reviewer_crnT · 2023-08-12
> >
> > Thanks a lot for your clarification, I have read your response.

---

### Official Review · Reviewer_vDxD · 2023-07-08

**Soundness:** 2 fair
**Presentation:** 2 fair
**Contribution:** 2 fair
**Rating:** 3
**Confidence:** 5

**Summary:**

The paper presents a new algorithm for learning rule ensembles and claims that these are interpretable, but does not present any support.

**Strengths:**

The proposed method is reasonable, but, in the context of other work in this area, not ground-shaking. The experimental evaluation is done well, but does not touch on interpretability.

**Weaknesses:**

There is no evidence that the learned rule sets are interpretable.
Rule complexity has not much to do with cognitive complexity.
The efficiency of algorithm is over-stated in the paper.

**Questions:**

None.

**Limitations:**

This is another paper that claims that rule ensembles are interpretable. No evidence is presented to that end, the claim is just derived from the fact that rules are, by themselves, interpretable. However, for example, random forests are well-known to be not interpretable, and they are also just a rule ensemble. In addition, the situation is even worse here, because in a random forest at least each individual rule is interpretable, and may be viewed as an explanation for all the examples it covers. In an additive boosting setting, this property also does not hold, because each rule corrects and refines predictions of previous rules, so rules can no longer be interpreted in isolation, but only in the context of all previous rules. Even a single example can not be easily explained by a gradient-boosted rule set, because one would have to understand the interaction of multiple rules.

The authors use the term "cognitive complexity" for something that is essentially the size of a rule set. Again, this is a complete misnomer, as the cognitive effort to parse a rule set does not only depend on the size of the theory. As explained above, there might be dependencies between rules, or rules may be considered in isolation (the latter having a much lower cognitive complexity). There are also factors such as the familiarity with the used concepts. For example, the cognitive effort required to read a page of text in your mother tongue is much lower than the cognitive effort required to read a page in a language that you are just learning, even though both, the content, as well as the syntactic length (essentially the author's measure of cognitive complexity) is the same.

It is a pity that they authors make such unfounded claims about intepretability, where they could simply present their work as an attempt to learn a simpler rule ensemble. As such, the work is reasonable, but also not great break-through. What they essentially propose (following previous work) is to re-optimize all weights once a new rule is added, and build an efficient algorithm around that idea. It gains a little in performance, as can be expected, but it is not great break-through.

The small advantage seems to be bought with an increase in computation time, which the authors interpret as "in the same order of magnitude" except for one case, where it is by a factor 26 slower. Actually, it seems to be the case that in most of the datasets, the algorithm is at least a factor of 2 smaller, sometimes worse.

Minor comments:

Some of the numbers in Table 1 are obviously wrong (e.g., testing risks of 109.5 or 4.115 for XGB).

---

> ### Author Rebuttal · Authors · 2023-08-09
>
> As stated in the global rebuttal, we take all the concerns regarding the nomenclature around interpretability and the term cognitive complexity serious, and we are happy to modify the language to reflect that gains in ensemble simplicity do not necessarily imply an interpretable model in absolute cognitive terms. As a whole, however, we believe your review misrepresents the actual core claims of the paper and does not recognise its main contributions.
>
> Firstly, the paper explicitly agrees with your assessment that not all rule ensembles can be interpretable for the simple reason that they can be too long to ever be parsed by one or even a group of interpreters. This is why the paper is based on the widely accepted premise that, all other cognitive factors being equal, shorter rule ensemble are relatively more interpretable than longer rule ensembles (alternatively, if one wants to insist on a dichotomous notion of “interpretability”, one could say “stand a better chance of being interpretable“).
>
> Secondly, based on that motivation the paper proposes a novel and technically non-trivial algorithmic approach to improve the trade-off of the ensemble size and the predictive performance. This approach is based on the idea of a weight correction step after each boosting round, which, as you point out, is described elsewhere. However, the main technical contributions of our work, which you do not mention, are then
>
> 1. to derive from this idea a novel sound objective function that successfully identifies the optimal rule to add when anticipating the subsequent weight updates (Theorem 4.3) and
> 2. to derive an efficient algorithm for optimising this objective function over incremental sequences of cut points, as it is required to practically use an objective function in either greedy or branch-and-bound rule learning.
>
> Both of these are original and non-trivial. Moreover, we demonstrate clearly that this approach allows the learning of rule ensembles with a better complexity / accuracy trade-off. While you assess the margins of improvement on average as not “ground-shaking” they show consistently over a wide range of settings, and, if you look, e.g., at the specific examples shown in Figure 1 of the paper or the newly uploaded figure in the global rebuttal, the improvements can be quite impressive for individual prediction risk levels.
>
> Finally, we would like to defend our statements regarding the efficiency of the algorithm. Firstly, a factor of two can be considered a price easy and worth to pay in scenarios where simpler rule ensemble are desired (it is therefore typically regarded as the same order of magnitude). Moreover, algorithmic details and certainly the implementation of newly published approaches tend to be suboptimal at first publication and are subsequently improved in an incremental fashion. Finally, in the theoretical asymptotic sense the proposed algorithm can certainly be considered efficient relative to straightforward solutions. Through Contribution 2, it enables all cut point evaluations for one input variable in the main loop with a time complexity growing only linearly in $n$ as opposed to quadratically, which one would achieve with a naive implementation.

---

> > ### Comment · Reviewer_vDxD · 2023-08-11
> >
> > I think you are missing the point. Additive rule ensembles by itself are not easily interpretable regardless of their complexity. Take a simple problem: We have one real-valued attribute in the range [0,10) and want to learn the rule for trunking off the digits after the commas.
> > A conventional rule learning algorithm learns:
> > * IF x < 1 then 0
> > * IF x in [1,2) then 1
> > ...
> > * IF x > 9 then 9
> >
> > You get the idea.
> >
> > Now what does an additive rule ensemble learn? Something like
> > * IF true then 5
> > * IF x in [0,5) then subtract 2
> > * IF x in [5,10) then add 2
> > * IF x in [0,2) and x in [5,7) then subtract 1
> > * IF x in [2,5) and x in [7,10) then add 1
> >
> > etc.
> >
> > The complexity of both rule sets is similar (in the same order of magnitude, if you want), but the first one is easily interpretable, but the second is not. You can construct similar examples with classification as well. The point is that in conventional rule sets, a single rule can be used for an explanation, whereas in additive rule ensembles every rule depends on all other rules. The number of rules and thus the complexity of the learned set or ensemble is not that important.
> >
> > Having said that, I repeat that I don't think that your contributions are the main problem, it is the way you present them. I would have liked your paper much better if you were not trying to oversell its contributions. This concerns not only interpretability, but also run-time. I agree that others such as the ones you cite do equate interpretability with complexity, but many of them learn rule sets, not additive ensembles, and (without checking) I dare say that only few actually dared to call syntactic complexity "cognitive complexity". In any case, if you research the literature on interpretability you find several papers that have in user studies obtained the correlation between interpretability and low complexity is only weak, often non-existing, or even negative. As I wrote, if you had simply put your paper as an approach for learning smaller ensembles, this would be quite o.k.
> >
> > The same with the run-time complexity. If you would have argued in the paper that a disadvantage of a factor of 2 is maybe not that bad, I would certainly not have complained, but glossing over such a difference with "in the same order of magnitude" is overselling.
> >
> > Try to analyze your algorithm objectively, and reviewers like myself would certainly be more positive. We all know that you cannot develop an algorithm that beats everything else in all regards, which makes papers that claim to have succeeded already suspicious. Even algorithms that are worse than others can be valuable contributions because, as you write, once published, an interesting idea can be picked up and improved by other researchers. But the important part is that you do a honest evaluation and not try to sell it with unfounded claims.
> >
> > I'm not sure what the discussion will yield, and what the handling chair will eventually decide, but I would certainly rather have a re-review of a thorough revision of this paper at another conference than to accept it here. Should it be accepted nevertheless, I do hope that you tone down you claims accordingly.

---

> > > ### Author Response · Authors · 2023-08-15
> > >
> > > Dear reviewer,
> > >
> > > You say again that the contributions of the paper are not the problem but that your reason for rejection is that they are “oversold” in a way that can only be addressed by a thorough revision. We agree that some formulations should be changed, but, on inspection of the paper, find it hard to see why this cannot be done with a couple of specific adjustments. Further, we had no intention of overselling results or to claim that, as you suggest, the proposed algorithm “beats everything else in all regards”, as indicated by our rather measured conclusion (l323) that "the proposed [...] approach *is a worthwhile alternative to previously published boosting variants for rule learning*, especially when targeting a beneficial risk-complexity trade-off and an overall small number of rules."
> > >
> > > We below provide a list of implicit and explicit claims made in the paper about its contributions that might have created the impression of “overselling”. For each of them, we give a justification or a simple reformulation that we intend to apply and we hope addresses your concerns. We hope that revisiting these points will lead you to a more favourable evaluation of our work:
> > >
> > > 1. The title “Orthogonal Gradient Boosting for Interpretable Additive Rule Ensembles”. We now understand that using the term interpretable especially in absolute terms might raise wrong expectations. Therefore we intend to use instead: “Orthogonal Gradient Boosting for Simpler Additive Rule Ensembles”. Similar we intend to change other mentions of the absolute term “interpretable”.
> > > 2. Using the term “*cognitive* complexity in terms of the number and length of rules” in the introduction and elsewhere. We understand that this term might raise a wrong impression and therefore intend to simply use “complexity” throughout the paper. We originally used the qualifier *cognitive* simply to differentiate it from "computational complexity", which is also sometimes considered for rule ensembles. Also note that the paper gave the definition as "in terms of number and length of rules" and therefore was clear about the intended meaning.
> > > 3. The explicit contribution statement that says that we "develop the first rule boosting algorithm that consistently optimizes the accuracy/complexity trade-off" and "derive a corresponding efficient algorithm for cut-point search”. We still consider both statements appropriate, as the novel objective function provably takes the weight correction into account and therefore increases the risk reduction per rule, and the cut point search algorithm is theoretically efficient, as its cost only depends linearly on the data size instead of quadratically as does the naive implementation.
> > > 4. The summary of the evaluation that says that the "computational cost remains *comparable* to the previous branch-and-bound learner". That point was also reinforced in the evaluation section where it was said that the new algorithms has "costs in the same order of magnitude" as XGB "except for one extreme case". While formulations like “comparable” and “same order of magnitude” are not uncommon to describe differences of a factor that is closer to 2 than to 10, we agree that these are at least ambiguous and can raise a wrong impression. We therefore intend to replace this by a more explicit summary of "an only modest computational overhead of a factor of about 2 for 26 of 34 datasets and a factor of less than 5 for seven of the remaining eight datasets."
> > >
> > > Please let us know if there are other formulations that you think are problematic and we are happy to address those as well.
> > >
> > > On a separate note let us comment on your general concerns about additive rule ensembles. Your example is indeed a nice benchmark problem where gradient boosted additive rule ensembles might not be the optimal choice. However, note that the additive rule ensemble that you suppose would be learned does not seem correct (rules 4 and 5 given have an empty coverage, perhaps you intended to use an “or” instead, but that is not something we consider). More generally, the class of additive rule ensembles contains the ideal ensemble that you sketch, and, depending on the employed approach, gradient boosting, or more likely column generation, might learn it. That said, we agree that GB will likely learn overlapping rules instead of the non-overlapping ones that are ideal in this scenario. If we know that this constraint is desirable, simply using rules from a single decision tree is indeed a better approach. However, there are lots of prediction problems where there are several independent additive effects, and in these cases overlapping rules are much more appropriate. Take for example risk factors of coronary heart disease. In these cases we would like to see two rule (e.g., +2 if smoker, +2 if obese) refining the general population risk log odds rather than listing all non-overlapping alternatives. Gradient boosting would be well equipped to produce exactly that.

---

> > > > ### Comment · Reviewer_vDxD · 2023-08-21
> > > >
> > > > Thanks for your comments.
> > > >
> > > > Your application scenario for, e.g., coronary heart disease, certainly makes sense, but maybe only if the learned weight changes are all integer-valued as in your example, in which case you have what is called a scoring system (Rudin has a few papers on that). With real-valued weights, I still think that these are difficult to interpret.
> > > >
> > > > In any case, the changes you sketch appear reasonable, and I would have been more positive on a version of the paper that does these.
> > > >
> > > > I am not changing my score, as I think these are too many and too substantial changes to do, but if the area chair is willing to take them on face value that's fine with me.
> > > >
> > > > I would, nevertheless, still not be excited about the paper (borderline at best), as I don't think that the proposed techniques are that great a break-through, as I wrote in my review. However, if the area chair and my fellow reviewers think otherwise, I will not block a possible acceptance of the paper (I am certainly in favor of having more rule learning papers at NeurIPS).

---

### Official Review · Reviewer_8EKB · 2023-07-10

**Soundness:** 3 good
**Presentation:** 3 good
**Contribution:** 3 good
**Rating:** 5
**Confidence:** 3

**Summary:**

This paper introduces Fully-Corrective Orthogonal Gradient Boosting (FCOGB), a novel algorithm aimed at facilitating interpretable rule learning. The study contends that existing rule learning algorithms often yield complex models that pose challenges for interpretation. FCOGB addresses this concern by generating simpler and more easily understandable models.
FCOGB is an extension of the widely employed gradient boosting algorithm, utilized for constructing predictive models. It employs a branch-and-bound search algorithm to identify the optimal set of rules that minimize prediction errors.

**Strengths:**

The proposed method is supported by theoretical justifications and intuitive explanations using figures. Additionally, the paper proposes algorithms with computational complexity analysis to efficiently implement the method, demonstrating practical applicability.

**Weaknesses:**

Despite an increase in the required training time (takes several times longer computation), the generalization performance does not improve. If this weakness is addressed, I believe it would become a very strong paper.

(Minor comments)
- Despite Figure 2 being referenced on page 5, the figure is actually inserted on page 3.
- The scatters plot in Figure 3 are difficult to interpret due to overlapping points. Please set alpha (transparency).

**Questions:**

- Table 1 evaluates the risk for cognitive complexity values ranging from 1 to 50 and provides the averaged risk. However, I'm wondering if there are any trends or patterns in the variations. For example, do the superior algorithms change when cognitive complexity is fixed at 5 compared to when it is fixed at 25? Additionally, if the range is changed to 1 to 25 or extended to 1 to 100, would the results be different? I believe this perspective would enhance the sense of conviction, and it might provide an approach for improving test error in your method.

- Are there any test error improvement strategies that can be considered? Otherwise, from the perspective of interpretability, are there cases where it is desirable for only the training error to decrease without an improvement in generalization performance (overfitting)?

- While SIRUS is presented as state-of-the-art, as seen in Figure 1, its performance looks unsuitable as a comparative reference. Do you have any idea why SIRUS does not work well in this example?

---

> ### Author Rebuttal · Authors · 2023-08-09
>
> We thank you for your thoughtful comments and hope you will consider upgrading your evaluation in the light of the following clarifications.
>
> **Generalization performance / test risk**
>
> You identified an unsatisfactory performance in terms of the test risk as the main weakness of the paper and assessed that, if addressed, "it would become a very strong paper". *As the results uploaded with the global rebuttal show, the test risk performance can in fact be fixed easily by introducing regularization to all the boosting variants.* Specifically, we now performed L2-regularization where the regularization hyper-parameter is tuned by an internal 5-fold cross-validation on the training set (and that tuning is performed for each of the targeted number of rules individually). This results in the proposed method having the best average test risk (over all considered complexity levels) for 23 of 34 datasets and the advantage compared to all other boosting variants to be statistically significant (for both train and test with appropriate Bonferroni correction). It is sensible that the proposed method benefits the most from regularization, given that it fits the training data most tightly for a given number of rules and is therefore, without regularization, more prone to overfitting for the larger considered cognitive complexity levels.
>
> **Effect of changing the considered complexity horizon**
>
> As you indicate, this question is related to the overfitting/regularization issue: in terms of training risk, the proposed method which fits the data tightest per rule comes up on top over a range of maximal complexity levels. In contrast, in terms of test error, it can backfire and, without regularization, enter the overfitting regime before other methods. However, that depends on the individual dataset. In the two examples in Figure 1, the test advantage of the proposed method is actually largest when going up to complexity 100. To focus on smaller and thus more interpretable complexity levels, as well as to speed up the overall computation time of the experiments, we originally decided to focus on 50. However, we appreciate the idea of including the results for the other maximal values and will do so in the revised supplementary information.
>
> **Test error improvements / relevance of the training error**
>
> As stated above, simple L2-regularization already achieves the desired results. Further improvements can likely be achieved by designing regularization terms that specifically punish longer rules. We focussed originally on the training error because it is a measure of how well the various objective functions approximate the “idealized boosting objective” of finding the single rule that allows to maximally reduce the training risk, and we believe that this is important in its own right. Notably, the advantage of the proposed method is retained when considering the regularized training risk instead (as reflected in the uploaded results in the global rebuttal).
>
> **Value of SIRUS baseline**
>
> Note that SIRUS is included as the state-of-the-art “generate-and-filter” method, and it was previously shown that gradient boosting tends to be superior to that approach. Hence, the main baselines are the previously published gradient boosting variants. We believe that the main reason for the problems of generate-and-filter, especially based on random forests, is that the individual rules present in the pool are not generated with the purpose of being good models in the context of small rule ensembles. Therefore there is a tendency that many of them are required to achieve a reasonable predictive performance.

---

> > ### Comment · Reviewer_8EKB · 2023-08-14
> >
> > Thank you for your updates.
> > Given the improvement in generalization performance, I would like to change the score from 4 to 5.

---

### Official Review · Reviewer_KXDe · 2023-07-10

**Soundness:** 2 fair
**Presentation:** 4 excellent
**Contribution:** 2 fair
**Rating:** 7
**Confidence:** 4

**Summary:**

This paper proposes a framework of fully corrective orthogonal boosting. The main algorithmic difference here is the objective for each next weak model. It is the cosine between the gradient (which is orthogonal to previous weak learners by construction) and the part of the new model that is orthogonal to previous models. Authors motivate their work by the need of interpretable models, so they restrict themselves to the case of rules as weak learners. Also, they use a variant of b&b algorithm for optimal weak learner search instead of commonly used greedy construction in depth.

Authors claim that this the paper proposes an algorithm for constructing shorter and more interpretable rules for Gradient Boosting of Decision Rules model. Experiments show that described method outperforms standard implementations of GB in case of using models of low complexity.

Although theoretical part is sound, practical questions are not thoroughly addressed or answered.

**Strengths:**

The main part of the paper is well-written, the terms, designations and ideas are clear. The proposed method is sound, reasonable and well described. The idea of orthogonal rule search in conjunction with fully-corrective goosting looks good. The theoretical part is described very well, the main formulations are correct, and the obtained contributions look important and are novel to the best of my knowledge. The proposed algorithm is justified and has the potential to compete with SOTA in the outlined formulation that refer to "cognitive complexity".

The main part of the paper is well-written. The terms, designations, and ideas are clear.

The only point I did not buy is the Poisson loss defined in line 109, in my opinion, incorrectly (or unclear), because, formally, from that definition, its minimum is at $f(x_i)=0$ independently on $y_i$.

**Weaknesses:**

I have the following concerns about the research direction itself. Claimed advantage of ensembles of rules over ensembles of trees is their human interpretability. However, I cannot agree that the decisions a rule ensemble makes can be treated as interpretable. Particularly, I argue that in the domains where interpretation is important summation of even two terms is usually not interpretable for humans. Most critical decisions in such domains like medicine and justice, partly science and risk management are usually based on several binary factors, not a sum of dozens of rules. Where ensembles of rules are really used in practice?

Second, I am disappointed that the term "cognitive complexity" was left without any background. I would expect references to some papers using this metric or explicit statement that this way to estimate models' complexity is originally proposed in the current paper. Futhermore, I would expect some consideration of actual research in psychology domain that address the problem of cognitive complexity of calculations.

For example, we can see in "Human knowledge models: Learning applied knowledge from the data." Plos one, 2022, by E. Dudyrev et al., that a human decision is usually based on:
-	Boolean operators: OR, AND, NOT, and thresholded Boolean SUM (arithmetic sum of
Boolean variables, compared to an integer threshold)
-	At most four (Boolean) variables, where each variable is used at most once\

These ideas are rather far from the concept of sums of dozens of rules

See also:

Lemonidis C., “Mental Computation and Estimation: Implications for mathematics education research, teaching and learning”, 2015,

Marois R et al, "Capacity limits of information processing in the brain," Trends in cognitive sciences, vol. 9, no. 6, pp. 296–305, 2005

Nys J. et al, "Complex Mental Arithmetic: The Contribution of the Number Sense," Canadian journal of experimental psychology, vol. 64, no. 3, pp. 215–220, 2010.

At last, but not least, the experimental part spoils the impression of the work and requires improvements:

- First of all, I see no hyperparameter tuning step description (e.g. regularization terms for XGB, number of boosting rounds, length of decision rules) in the section on experiments. Are there any hyperparameters which may have a significant impact on the performance of FCOGB? Where they left "defaulted" or were they were tuned by a separate step of an algorithm?

- In the beginning of Section 5, it is mentioned that you use only 5 runs for each dataset with < 50 cognitive complexity (CC) limit. But then I see averaging over complexities between 1 and 50 in the description. What does it mean? I suppose that CC may alter in different runs but it is limited to 50, is it true? Or did authors perform exhaustive search of all possible CCs and averaged over them? If the first is true, I have a doubt that different model may have had different mean CC values, so that it is not quite fair comparison results. Is the second is true, then it is unclear why such an averaging can prove something

- It would be interesting to see the dynamics of quality with respect to increasing CC. In particular, some graphs that plots quality vs CC to see which algorithm uses the CC limit more effectively.

- It may be useful to provide comparison with other variants of fully-corrective boosting implementations since the quality gain may origin from described by this scheme only

- Time limitations should be discussed more in terms of time per CC point and pareto curves (time to achieve the desired quality)

- How should we interpret relatively low quality for regression problems?

- This paper addresses interpretability of trained decision rules, so it would be profitable to demonstrate a difference in the simplicity of interpretation for FCOGB rules and, e.g., XGB rules

**Questions:**

- Equation (4) goes far from original GB. The point $(Q_{t-1};g) \alpha^*$ in the functional space is used as a target for the next step of rule search. You can go one step further this way: use the strict minimizer $f^*$ of the loss functional as a target instead of $(Q_{t-1};g) \alpha^*$. This looks a simpler choice, which can also increase boosting convergence and thus simplicity of the trained ensemble. Did you consider this way?

- How b&b approach to rule search matches previous literature on rule gradient boosting?

**Limitations:**

I do not see any particular limitation of the proposed work

---

> ### Author Rebuttal · Authors · 2023-08-09
>
> We thank you for the constructive feedback and overall positive assessment. Please find below point-by-point responses of your concerns and questions.
>
> **Usage of the terms interpretability and cognitive complexity**
>
> We understand your comments regarding the interpretability of rule ensembles and are happy to modulate the language around this notion (see global rebuttal). We do maintain however that, all other cognitive variables being equal, shorter rule ensembles are relatively more interpretable than longer rule ensembles, which is in line with many published works in the rule learning community (e.g., Refs 2, 3, 6, 10, 14, 29). We invite you to inspect again the examples shown in Figure 1 as a support of this intuitive claim as well as the pdf attached to the global rebuttal. Moreover, we would like to clarify that the employed complexity measure was already used in other work (Refs 2 and 6). We used the term “cognitive complexity” in our work mainly to disambiguate the concept from “computational complexity”, which can also be considered a complexity measure for models (in terms of the expected number of iterations one has to carry out to compute a prediction). As expressed in the global rebuttal, we are happy to change the wording to avoid misleading conclusions. Moreover, we plan to point to the provided references to highlight the difference in focus to work in cognition science.
>
> **Description of hyper parameter tuning**
>
> In the submitted version we aimed to keep experiments as simple as possible and avoided hyper-parameters by not applying any regularization or upper limit for the rule length. For the number of rules we simply ran all boosting variants until the designated complexity level was reached and then considered all intermediate complexities of rule ensembles that can be produced by a method (see also clarification below regarding the complexity levels). As described in the global rebuttal, not using regularization kept the focus mainly on the training performance that could be achieved with a certain complexity and inevitably led to overfitting for the largest complexity levels. Based on the reviewer comments we have since incorporated regularization into all boosting methods by choosing lambda values from a fixed grid via performing an internal 5-fold cross-validation on the training set. Indeed with this modification the results are much more favorable for the proposed method in terms of test performance (see pdf in global rebuttal).
>
> **Clarification of complexity levels**
>
> Ensembles for all complexity levels are considered that a method can produce (up the maximum considered). To realize the different ensembles the number of rules is varied. Indeed, for different repetitions based on different training/test splits, the achieved complexities and risk levels differ. Hence we consider average complexity / risk trade-off curves. As the exemplary curves shown in Figure 1 demonstrate, usually risk advantages are achieved on a wide range of complexity levels (nb we believe that these are the curves that you ask for). However, to obtain a simple quantitative summary per dataset we then consider averaged risks overall complexity levels, which leads to the overall statistical comparison between methods.
>
> **The performance for regression versus classification problems**
>
> The achieved risks for regression are typically lower than those for classification for the considered complexity levels. We believe this is due to a better alignment of the boosting objective with the actual error minimizing direction due to the simpler squared loss when compared with the logistic loss (which requires many iteration even when exact gradient descent can be performed, which is not the case for gradient boosting where we approximate search directions with the available rules).
> The possibility of using the strict minimiser in Equation 4
>
> This would indeed be the optimal rule to add and correspond to what we can refer to as “idealized boosting” with weight correction (see also Ref 26), which considers this variant for theoretical investigation. However, note that in practice we have to find the optimal rule by implicitly searching the space of all rules and it is completely unclear how to perform this optimization efficiently for this idealized objective. In fact, all the various boosting objectives can be considered approximations to this ideal objective, and we can see based on the training performance that, among the known choices, the function proposed in our work is the best such approximation.
>
> **Branch-and-bound rule search in previous literature**
>
> Branch-and-bound search have been considered for the simple objectives in separate-and-conquer rule learners (see 11) and subgroup discovery (see 15). In the context of gradient rule boosting, to our knowledge, using branch-and-bound search was proposed in Ref 3, however using the XGB objective function. In our work we adopt it to the refined objective function. However, we should note that our technical contributions go beyond the usage of branch-and-bound, as the proposed objective function can also be optimized with the more typical greedy algorithm and the fast incremental computations provided in our paper are also necessary in this approach.
>
> **Other points**
>
> We appreciate the other suggestions and will provide more detailed comparison curves in the supplementary material as well as an ablation study to show the impact of individual aspects of the proposed methods, in particular the performance when using the faster greedy algorithm for single rule optimization. Finally, there was indeed a typo in the Poisson loss. The correct definition is $l_{poi}(f (x_i), y_i) = y_i\log y_i − y_i f(x_i) − y_i + \exp(f(x_i))$.

---

### Author Rebuttal · Authors · 2023-08-09

We thank all reviewers for their thoughtful comments and their mostly positive assessment. In addition to the individual rebuttals, we would like to clarify here two central points:

1. The test performance of the proposed algorithm and the fact that it can be easily improved by regularization (see updated results attached).
2. The main contributions of the paper and how they are related to interpretability .

Regarding the test error comparison we are happy to report that the proposed method in fact significantly outperforms all baselines when performing adequate regularization. In the originally submitted experiment we omitted regularization in an effort to keep the workflow as simple as possible and to focus on the direct effect of the new objective function on the training error. However,  this led to sometimes large degrees of overfitting especially for the proposed method, which, due to the improved objective function, generally fits the training data tighter for a given number of rules. When introducing L2-regularization to all boosting variants and choosing the corresponding lambda value by 5-fold cross validation on the training set, the proposed method has the best test performance on 23 out of 34 datasets, and a Bonferroni corrected t-test shows that the improvement over all other boosting variants is statistically significant for both train and test error. Please see the table in the attached pdf for detailed results.

Regarding the notion of interpretability, we would like to acknowledge that claims regarding interpretability in absolute terms may indeed require considerations of cognitive psychology and involve the familiarity of the investigator with symbols among other things. We are happy to modulate the language in the paper accordingly. That being said, we do maintain that, all other cognitive variables being equal, shorter rule ensembles stand a better chance of being interpretable than longer rule ensembles, which is in line with lots of work published in the rule learning community (e.g., Refs 2, 3, 6, 10, 14, 29). We believe this to be also self-evident in the examples of concrete rule ensembles presented in Figure 1 of the paper as well as in Figure 1 of the pdf attached to this rebuttal. These cases demonstrate that the proposed method produces much shorter and easier to interpret results in several concrete examples, which is in addition to the positive large-scale quantitative evaluation that has been performed on a wide range of prediction tasks.

These advantages are enabled by the two main technical contributions of the paper: a novel objective function that identifies optimal rule bodies to add when taking a subsequent weight correction step into account as well as in an efficient algorithm to compute this objective function for an incrementally increasing collection of data points. The first allows a more stringent optimization of the length versus accuracy trade-off of rule ensembles. The latter is crucial to practically employ the new objective function in branch-and-bound as well as greedy rule search.

---

### Decision · Program_Chairs · 2023-09-21

**Decision:**

Reject

**Comment:**

The paper proposes a new framework of fully corrective orthogonal boosting, whose claimed advantages over standard gradient boosting is to provide smaller and more interpretable rule ensembles.

Reviewers were divided on this paper. They all seem to appreciate the originality and the soundness of the proposed algorithm. Reviewers KXDe and vDxD had concerned about the misuse of the term interpretability. I tend to agree with them that the paper oversells its contribution as a means of improving interpretability, when it is more a way to build less complex additive rule ensembles. I think the authors need to incorporate the changes that they promised in their responses in subsequent version of their paper.

Two reviewers mentioned that the experimental results were not very convincing because the generalization performance of the method was not better than that of other competitors. In the submitted paper, there is indeed no very strong evidence that the approach provides a better risk/complexity tradeoff than other methods. Following a suggestion of one of the reviewers, the authors rerun all experiments by tuning the regularization parameter, which now makes their method more competitive. In their responses, they explain this improvement by the fact that their method suffers more from overfitting than the other boosting variants. This is a nice improvement, but I think this new observation changes the discussion of the experimental results a bit too much compared to the original paper.

Given the required changes (in terms of presentation and discussion of results), I would be more comfortable reviewing a new version of the paper rather than accepting it as is. For this reason, I recommend rejection.